

# The relationship between the severity and complications of Henöch-Schönlein purpura in children and dietary inflammatory index: a retrospective cohort study

Jinshu Chen[1], Pihou Chen[2], Yijin Song[1], Jiaxin Wei[1], Shiya Wu[1], Fan Wu[1] and Zhiquan Xu[1]

[1] Department of Rheumatology and Immunology for Children's Nephropathy, Hainan Women and Children's Medical Center, Haikou, China
[2] Department of Children's Rehabilitation, Hainan Women and Children's Medical Center, Haikou, China

Corresponding author
Zhiquan Xu, cjs20211022@163.com

## ABSTRACT

**Purpose:** To investigate the association between the Dietary Inflammatory Index (DII) and disease severity as well as complications in children diagnosed with Henöch-Schönlein purpura (HSP), shedding light on the potential influence of dietary factors on HSP.

**Methods:** A retrospective cohort study was conducted, enrolling children aged 2–14 years diagnosed with HSP. Participants were divided into low and Pro-inflammatory dietary groups based on their DII scores. Biomarkers, nutrient intake, blood lipid profiles and disease complications were compared between the two groups. Spearman correlation analysis was performed to assess the relationship between DII and complications.

**Results:** A total of 115 patients, including 56 patients with anti-inflammatory dietary and 59 with pro-inflammatory dietary, were included. The pro-inflammatory dietary group demonstrated significantly elevated of C-reactive protein, tumor necrosis factor-$\alpha$, interleukin-6, erythrocyte sedimentation rate, white blood cell count, eosinophils, IgE, consumption of total calories, protein, carbohydrates, fiber, fat intake, total cholesterol, LDL cholesterol, HDL cholesterol, triglycerides, VLDL cholesterol, complications of renal, skin, gastrointestinal, coagulation and respiratory in comparison to the anti-inflammatory dietary group. DII was positively correlated with renal, skin, gastrointestinal, coagulation and respiratory complications.

**Conclusion:** The study highlights the potential influence of dietary inflammatory potential, as quantified by the DII, on disease severity and complications in children with HSP. Understanding the interplay between dietary patterns and inflammatory responses in pediatric vasculitis has implications for the management of HSP, emphasizing the relevance of considering dietary interventions to optimize clinical outcomes and improve the overall well-being of affected children.

## INTRODUCTION

Henöch-Schönlein purpura (HSP) is a systemic vasculitis characterized by the deposition of immune complexes containing predominantly IgA in small vessels throughout the body, leading to inflammation and tissue damage (*Al E'ed, 2021*; *Ma et al., 2021*; *Leung, Barankin & Leong, 2020*). The classic tetrad of HSP encompasses palpable purpura (small, red to purple skin spots that are palpable), arthritis or arthralgia, gastrointestinal symptoms (such as abdominal pain, vomiting, and gastrointestinal bleeding), and nephritis (manifesting as hematuria and proteinuria) (*Reamy, Servey & Williams, 2020*; *Urganci et al., 2022*; *Di Pietro et al., 2019*).

HSP primarily affects children, with the majority of cases reported in those between 2 and 6 years of age (*Di Vincenzo et al., 2023*; *Zumbro et al., 2023*; *Zhong & Ding, 2023*). While the exact etiology of HSP remains incompletely understood, it is believed to involve an aberrant immune response triggered by various environmental factors, infectious agents, and genetic predisposition (*Di Pietro et al., 2019*; *Hammad et al., 2023*; *Maritati et al., 2020*). The clinical course of HSP is highly variable, ranging from a self-limiting and benign condition to severe and persistent disease with renal involvement and long-term complications (*Reamy, Servey & Williams, 2020*; *Oni & Sampath, 2019*; *Sestan & Jelusic, 2023*).

The systemic nature of HSP, with its potential to affect multiple organs and systems, underscores the significance of understanding the factors that may modulate disease severity and outcomes (*Limavady & Suryana, 2022*). The association between dietary factors, systemic inflammation, and immune dysregulation has garnered increasing attention in the context of various inflammatory conditions, raising the question of whether dietary inflammatory potential may contribute to the clinical manifestations and progression of HSP (*Leung, Barankin & Leong, 2020*).

In recent years, emerging evidence has suggested that dietary patterns and their inflammatory potential may play a significant role in modulating the immune response and disease outcomes in various inflammatory conditions, including vasculitis. The Dietary Inflammatory Index (DII) is a composite scoring algorithm developed to quantify the overall inflammatory potential of an individual's diet based on the intake of specific nutrients and bioactive compounds (*Li et al., 2021*; *Mohammadi et al., 2023*; *Motamedi et al., 2022*). A higher DII score reflects a more pro-inflammatory diet, while a lower score indicates a more anti-inflammatory diet (*Souza, Ferreira & Dos Santos, 2023*; *Vajdi, Farhangi & Mahmoudi-Nezhad, 2022*; *Hayati, Jafarabadi & Pirouzpanah, 2022*). Several epidemiological studies have demonstrated the association between high DII and an increased risk of chronic inflammatory conditions, such as cardiovascular diseases, metabolic syndrome, and certain cancers (*de Freitas et al., 2022*; *Farazi, Jayedi & Shab-Bidar, 2023*; *Syed Soffian et al., 2022*).

Despite the growing understanding of the potential impact of diet on systemic inflammation and disease susceptibility, limited research has focused on the role of dietary inflammatory potential in pediatric vasculitis, particularly HSP. Therefore, this study aims to investigate the association between DII and disease severity as well as complications in

children diagnosed with HSP. The findings may contribute to a deeper understanding of the interplay between dietary patterns and inflammatory responses in pediatric vasculitis, ultimately paving the way for novel dietary-based interventions in the management of HSP.

## MATERIALS AND METHODS

### Study subjects

This was a retrospective cohort study. The cohort included children 2 to 14 years of age who have received a diagnosis of HSP were recruited from the clinical data database of this hospital. The study period was from April 2021 to March 2024. The cohort was divided into two groups according to the dietary inflammation index score: low dietary inflammation index group and high dietary inflammation index group.

### Inclusion and exclusion criteria

**Inclusion criteria:** Children aged 2–14 years diagnosed with HSP (*Leung, Barankin & Leong, 2020*); availability of complete dietary records for a period of at least 6 months prior to the diagnosis of HSP; availability of follow-up data for at least six months post-diagnosis; parents or legal guardians were willing to provide informed consent for their children to participate in this study. This study was approved by the Ethics Committee of Hainan Women and Children's Medical Center and complies with the ethical guidelines of the Helsinki Declaration (No. HNWCMC-2021-38). This study obtained written informed consent from the patient's parents or legal guardians.

 Exclusion criteria: Children with co-existing significant chronic inflammatory conditions, such as inflammatory bowel disease, juvenile rheumatoid arthritis, or systemic lupus erythematosus, which may confound the relationship between dietary factors and disease severity in HSP; Children with a history of major dietary changes or modifications during the study period, which may substantially affect the accuracy and reliability of the dietary records.

### Data collection

Data on dietary intake will be collected through validated food frequency questionnaires and dietary records. Information on disease complications will be obtained from medical records, including laboratory results, clinical assessments, and diagnostic imaging reports. Other relevant demographic and clinical data will also be collected, including age, sex, comorbidities, and medications.

### Blood test

Prior to the test, subjects were required to fast for 12 h. In the morning, 3 ml of blood was drawn from the antecubital vein while the subject was in a fasting state (*Andreazza et al., 2019*). The blood was then subjected to heparin anticoagulation, followed by centrifugation to separate the plasma. An automated biochemical analyzer (Cobas c701; Roche Diagnostics, Basel, Switzerland) was used to measure the levels of C-reactive protein, tumor necrosis factor-$\alpha$, interleukin-6, erythrocyte sedimentation rate, white blood cell

count, eosinophils, IgE, serum albumin, vitamin D, thyroid-stimulating hormone, total cholesterol, triglycerides, low-density lipoprotein (LDL) cholesterol, high-density lipoprotein (HDL) cholesterol, very low-density lipoprotein (VLDL) cholesterol.

Biomarkers were selected based on their established roles in inflammatory processes and their relevance to pediatric vasculitis, particularly HSP. A panel of inflammatory biomarkers, including C-reactive protein (CRP), tumor necrosis factor-α (TNF-α), interleukin-6 (IL-6), erythrocyte sedimentation rate (ESR), white blood cell count, eosinophils, and IgE, was chosen due to their documented association with inflammatory conditions and vasculitis (*Li et al., 2021*). To assess the specificity of the observed associations and to serve as negative controls, we added biomarkers not typically associated with inflammation or HSP. These biomarkers were selected based on their lack of established roles in inflammatory pathways or vasculitis pathogenesis, including serum albumin, vitamin D levels, and thyroid-stimulating hormone (TSH), which are not expected to be significantly altered in HSP.

Quality control measures were rigorously implemented throughout the process of measuring lipid profiles and inflammatory biomarkers (*Andreazza et al., 2019*). Standard operating procedures (SOPs) were followed for all assays, and the laboratory personnel were trained and certified in performing these procedures. The Cobas c701 analyzer was calibrated daily using manufacturer-provided controls to verify the performance of the assays. The instrument's internal quality control (IQC) program was run concurrently with patient samples to monitor the precision and accuracy of the measurements. IQC materials included low, medium, and high controls for each parameter, covering the expected range of values in the study population. Any deviations beyond the predefined limits led to immediate corrective action and re-testing of the samples.

## DII

The participants' dietary intake was assessed using a food frequency questionnaire (FFQ) to investigate their food consumption over the past 3 months. Trained investigators conducted interviews to determine the types and quantities of foods consumed by the participants. Food portion sizes were calculated using household measures, food models, or food atlases. The intake of a specific food or nutrient was calculated based on the "Chinese Food Composition Table (6th edition)" and then used to calculate the individual's overall DII based on the average daily nutrient intake (*Marx et al., 2021*; *Nasab et al., 2023*). The DII was calculated according to the algorithm developed by Nitin Shivappa and colleagues at the University of South Carolina, which incorporates dietary data from 11 representative populations worldwide (*de Mello et al., 2023*). A total of 45 food parameters were assigned DII scores based on their potential to alter serum inflammatory markers, with a negative DII indicating anti-inflammatory potential and a positive DII indicating pro-inflammatory potential (*Hariharan et al., 2022*). Standardized scores (Z-scores) were calculated using the formula (*Shivappa et al., 2014*): (the daily intake of a particular dietary component or nutrient—the global average daily intake of that dietary component or nutrient)/ the standard deviation of the global average daily intake of that dietary component or nutrient. To minimize the influence of outliers and

right-skewed distributions, the Z-scores were converted into percentile ranks; in order to achieve a symmetrical distribution centered around 0, with −1 representing the maximum anti-inflammatory potential and +1 representing the maximum pro-inflammatory potential, each percentile rank was doubled, then reduced by 1, and finally multiplied by the inflammatory effect score for each dietary component. The sum of all food inflammatory effect scores yielded the DII for each participant (*Phillips et al., 2019*; *Vicente et al., 2020*).

For example: Vitamin C intake 100 mg a day, global average daily intake 80mg, the standard deviation of the global average daily intake of Vitamin C is 20 mg, the Z-score is (100-80)/20=1. The Z-score multiplied by the inflammatory effect score converted to a DII score. Similarly, the DII scores for the remaining food items and nutrients are calculated. The final DII score for the individual is obtained by summing all the individual DII scores for each food item or nutrient.

## Data analysis

The demographic and clinical characteristics of the cohort were analyzed using SPSS 25.0 statistical software. Descriptive statistics were presented for categorical data as ($n$(%)) when the sample size was ≥40 and the theoretical frequency (T) was ≥5. The chi-square test with the basic formula was used as the test statistic ($\chi^2$) for these cases. However, if the sample size was ≥40 but the theoretical frequency was $1 \leq T < 5$, the chi-square test was adjusted using the correction formula. When the sample size was <40 or the theoretical frequency (T) was <1, statistical analysis was performed using Fisher's exact test. For normally distributed continuous data, the mean and standard deviation were presented as ($\bar{x} \pm s$). For non-normally distributed data, statistical analysis was carried out after variable transformation to achieve normal distribution, and the t-test was used. The relationship between DII and complications was assessed using Spearman's correlation analysis, and statistically significant differences between the two groups were identified. A significance level of $p < 0.05$ was used to indicate statistical significance.

## RESULTS

### General information

In this study, we aimed to investigate the association between DII and demographic as well as clinical characteristics in the study population (Table 1). A total of 115 subjects were divided into an anti-inflammatory dietary group ($n = 56$) and a pro-inflammatory dietary group ($n = 59$). The analysis of demographic characteristics indicated comparable distribution with respect to age (8.62 ± 1.52 *vs.* 8.89 ± 1.75, $p = 0.387$), body mass index (17.98 ± 2.05 *vs.* 18.16 ± 2.22, $p = 0.65$), gender, disease duration (15.41 ± 6.83 *vs.* 16.75 ± 7.24, $p = 0.309$), HSP family history, physical activity (3.52 ± 1.04 *vs.* 3.21 ± 0.88, $p = 0.09$), and mode of delivery. The distribution of HSP family history (32.14% *vs.* 35.59%, $p = 0.846$) as well as physical activity (1.71, $p = 0.09$) did not show statistically significant differences between the two groups. Furthermore, no significant association was found between DII status and mode of delivery ($p = 1$). A significant difference was observed between the DII scores of the anti-inflammatory diet group and the pro-inflammatory diet

**Table 1 Demographic and clinical characteristics of the study population.**

| Parameters | Anti-inflammatory dietary group ($n$ = 56) | Pro-inflammatory dietary group ($n$ = 59) | t/X$^2$ | $p$ value |
|---|---|---|---|---|
| Age (years) | 8.62 ± 1.52 | 8.89 ± 1.75 | 0.869 | 0.387 |
| Body mass index (kg/m$^2$) | 17.98 ± 2.05 | 18.16 ± 2.22 | 0.455 | 0.65 |
| Gender (Male/Female) | 30/26 | 32/27 | 0 | 1 |
| Disease duration (months) | 15.41 ± 6.83 | 16.75 ± 7.24 | 1.023 | 0.309 |
| HSP family history | 18 (32.14%) | 21 (35.59%) | 0.037 | 0.846 |
| Physical activity (hrs/week) | 3.52 ± 1.04 | 3.21 ± 0.88 | 1.71 | 0.09 |
| Mode of delivery | | | 0 | 1 |
| Vaginal delivery | 27 (48.21%) | 29 (49.15%) | | |
| Cesarean section | 29 (51.79%) | 30 (50.85%) | | |
| DII scores | −5.15 ± 1.48 | 3.46 ± 1.23 | 33.998 | <0.001 |

**Table 2 Biomarkers in study participants in both groups.**

| Parameters | Anti-inflammatory dietary group ($n$ = 56) | Pro-inflammatory dietary group ($n$ = 59) | t | $p$ value |
|---|---|---|---|---|
| C-reactive protein (mg/L) | 3.21 ± 1.54 | 4.87 ± 2.11 | 4.85 | $p < 0.001$ |
| Tumor necrosis factor-α (pg/mL) | 12.64 ± 3.81 | 14.56 ± 4.26 | 2.541 | 0.012 |
| Interleukin-6 (pg/mL) | 7.77 ± 2.83 | 9.67 ± 2.94 | 3.547 | $p < 0.001$ |
| Erythrocyte sedimentation rate (mm/h) | 15.36 ± 4.56 | 17.88 ± 5.23 | 2.75 | 0.007 |
| White blood cell count ($10^9$/L) | 7.18 ± 2.24 | 8.74 ± 2.57 | 3.47 | $p < 0.001$ |
| Eosinophils (%) | 4.15 ± 1.23 | 5.37 ± 1.45 | 4.886 | $p < 0.001$ |
| IgE (IU/mL) | 236.78 ± 45.67 | 320.96 ± 58.21 | 8.651 | $p < 0.001$ |
| Serum albumin (g/dL) | 4.16 ± 0.85 | 3.97 ± 0.77 | 1.257 | 0.211 |
| Vitamin D (ng/mL) | 25.48 ± 2.47 | 23.75 ± 2.43 | 1.597 | 0.113 |
| Thyroid stimulating hormone (mIU/L) | 5.87 ± 0.59 | 5.68 ± 0.69 | 1.583 | 0.116 |

group (−5.15 ± 1.48 *vs*. 3.46 ± 1.23, $p < 0.001$). These findings suggest that there were no significant differences in demographic and clinical characteristics between the low DII and Pro-inflammatory dietary groups, indicating a well-balanced study population.

## Biomarkers

First, this study explored the effect of DII on inflammatory markers, and the results showed that in the pro-inflammatory dietary group demonstrated significantly elevated levels of C-reactive protein (4.87 ± 2.11 *vs*. 3.21 ± 1.54, $p < 0.001$), tumor necrosis factor-α (14.56 ± 4.26 *vs*. 12.64 ± 3.81, $p = 0.012$), interleukin-6 (9.67 ± 2.94 *vs*. 7.77 ± 2.83, $p < 0.001$), erythrocyte sedimentation rate (17.88 ± 5.23 *vs*. 15.36 ± 4.56, $p = 0.007$), white blood cell count (8.74 ± 2.57 *vs*. 7.18 ± 2.24, $p < 0.001$), eosinophils (5.37 ± 1.45 *vs*. 4.15 ± 1.23, $p < 0.001$), and IgE (320.96 ± 58.21 *vs*. 236.78 ± 45.67, $p < 0.001$) in comparison to the anti-inflammatory dietary group (Table 2). However, serum albumin, vitamin D, thyroid stimulating hormone levels had no significant difference between the two groups. These

findings suggest a strong association between dietary inflammatory potential and elevated inflammatory biomarkers, indicating a potential effect of diet on inflammatory status.

## Nutrient intake

Next, we analyzed the macronutrient intake of the two groups of patients. Compared with the anti-inflammatory dietary group, participants in the pro-inflammatory dietary group demonstrated significantly higher consumption of total calories (1,932.13 ± 221.03 *vs.* 1,753.23 ± 201.44, $p < 0.001$), protein (69.42 ± 12.45 *vs.* 61.14 ± 12.23, $p < 0.001$), carbohydrates (271.34 ± 35.14 *vs.* 243.12 ± 31.42, $p < 0.001$), fiber (31.24 ± 6.14 *vs.* 25.63 ± 5.86, $p < 0.001$) and fat intake (79.64 ± 18.32 *vs.* 72.45 ± 15.78; $p = 0.026$) (Fig. 1). These outcomes indicate distinct dietary patterns between the two groups, suggesting a potential link between DII and nutrient intake, which may influence inflammatory status as indicated by the previous inflammatory biomarker analysis.

## Blood lipid profile

In addition, the lipid profiles of the low DII and pro-inflammatory dietary groups were compared in this study. Subjects following pro-inflammatory dietary patterns exhibited significantly higher concentrations of total cholesterol (195.74 ± 25.64 *vs.* 172.46 ± 21.11, $p < 0.001$), LDL cholesterol (129.36 ± 21.37 *vs.* 106.13 ± 15.54, $p < 0.001$), HDL cholesterol (49.68 ± 11.36 *vs.* 42.97 ± 8.87, $p < 0.001$), triglycerides (126.98 ± 12.76 *vs.* 118.34 ± 15.36, $p = 0.001$), and VLDL cholesterol (27.64 ± 3.58 *vs.* 22.85 ± 3.55, $p < 0.001$) in comparison to those in the anti-inflammatory diet cohort (Fig. 2). These findings suggest a strong association between high DII and adverse blood lipid profiles, highlighting the potential impact of dietary inflammatory potential on lipid metabolism.

## Disease complications

Immediately following this, the analysis of the relationship between DII and disease complications in this study revealed significant differences between the two groups (Table 3). Individuals from the pro-inflammatory dietary group were observed to have considerably higher rates of renal complications (22.03% *vs.* 7.14%, $p = 0.047$), skin complications (30.51% *vs.* 10.71%, $p = 0.017$), gastrointestinal complications (25.42% *vs.* 5.36%, $p = 0.007$), coagulation disorders (32.2% *vs.* 8.93%, $p = 0.005$), and respiratory complications (22.03% *vs.* 5.36%, $p = 0.021$) when juxtaposed with the anti-inflammatory dietary group. Although the difference in neurological complications was not statistically significant between the two groups (18.64% *vs.* 7.14%, $p = 0.12$), the overall pattern suggests a potential association between high DII and increased prevalence of various disease complications. These results underscore the possible impact of dietary inflammatory potential on the development of disease complications, pointing to the importance of dietary factors in disease management.

## Correlation analysis

Finally, the correlation analysis of the DII with complications in children with HSP revealed significant positive correlations (Table 4). The DII demonstrated a moderate to

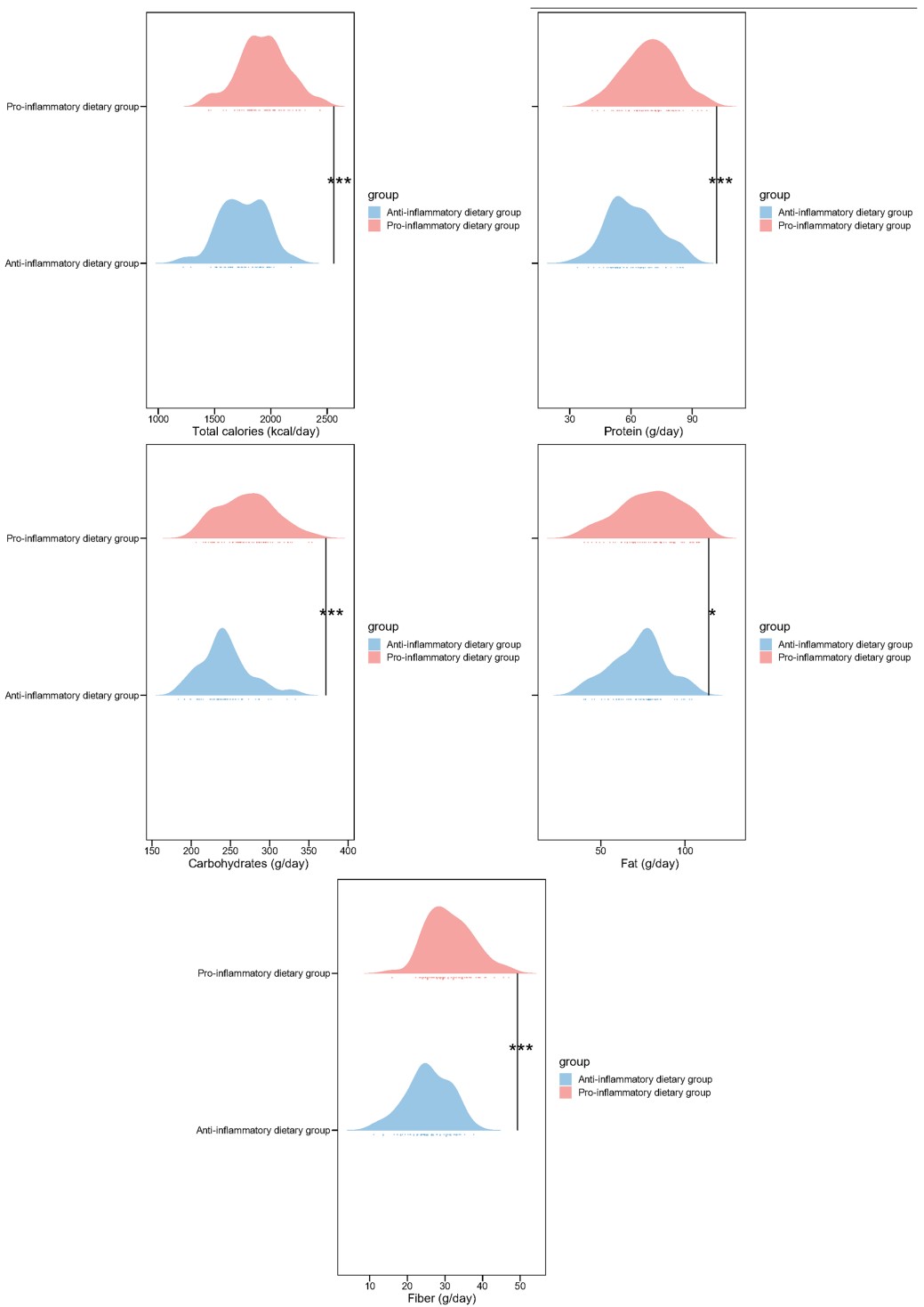

**Figure 1 Nutrient intake in study participants of two groups.** $^{*}p < 0.05$; $^{***}p < 0.001$.

strong positive correlation with various disease complications, including renal ($r = 0.21$, $p = 0.024$), skin ($r = 0.243$, $p = 0.009$), gastrointestinal ($r = 0.276$, $p = 0.003$), coagulation disorders ($r = 0.286$, $p = 0.002$), and respiratory complications ($r = 0.241$, $p = 0.01$).

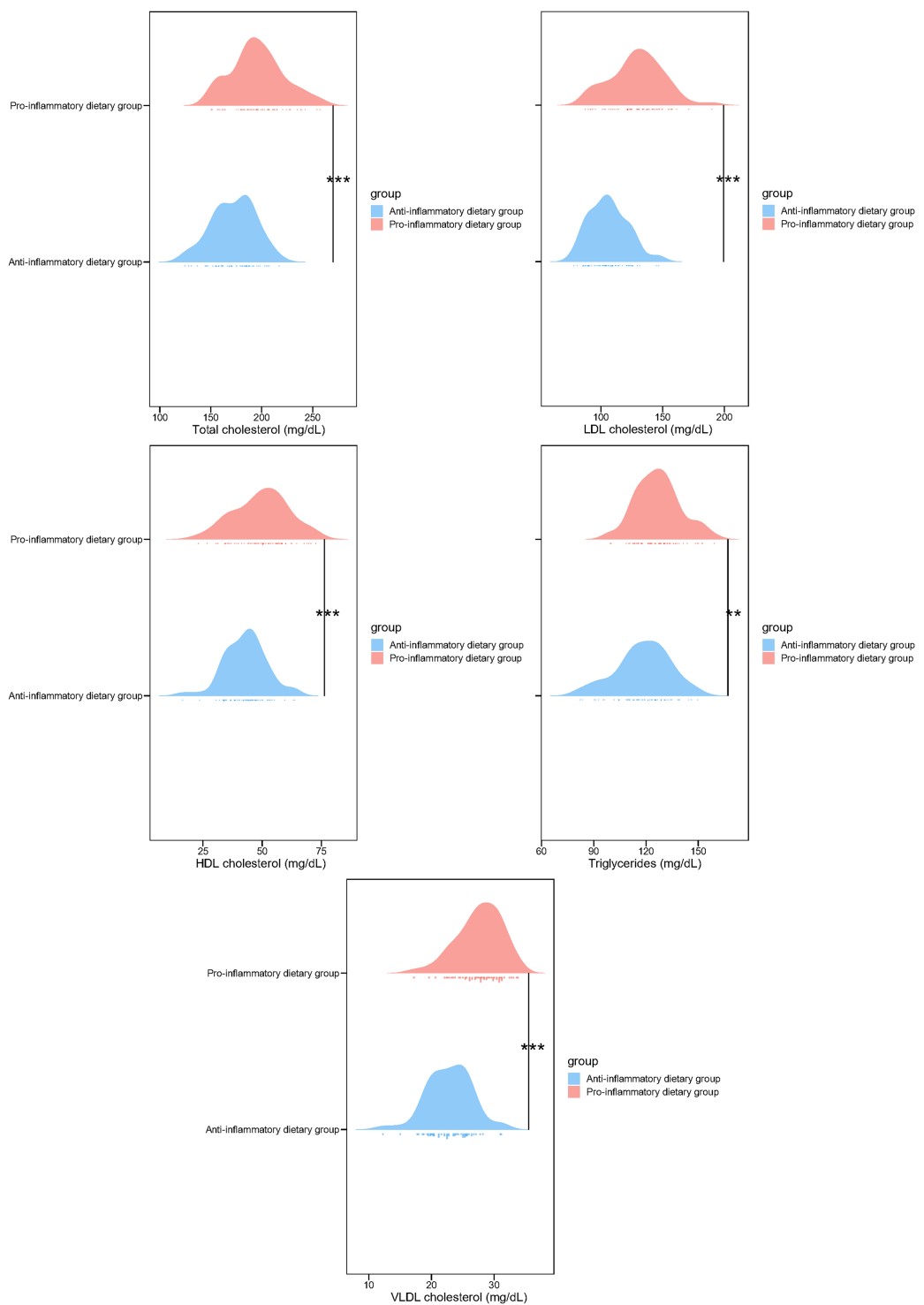

**Figure 2 Blood lipid profile in study participants of two groups.** $^{**}p < 0.01$; $^{***}p < 0.001$.

Although the correlation with neurological complications was not statistically significant ($r = 0.171$, $p = 0.068$), the overall findings indicate a notable association between higher DII and a higher risk of various complications in children with HSP. These results underscore

**Table 3 Disease complications in study participants of two groups.**

| Parameter (%) | Anti-inflammatory dietary group ($n$ = 56) | Pro-inflammatory dietary group ($n$ = 59) | t | *p*-value |
|---|---|---|---|---|
| Renal complications | 4 (7.14%) | 13 (22.03%) | 3.944 | 0.047 |
| Skin complications | 6 (10.71%) | 18 (30.51%) | 5.671 | 0.017 |
| Gastrointestinal complications | 3 (5.36%) | 15 (25.42%) | 7.309 | 0.007 |
| Coagulation disorders | 5 (8.93%) | 19 (32.2%) | 8.068 | 0.005 |
| Respiratory complications | 3 (5.36%) | 13 (22.03%) | 5.352 | 0.021 |
| Neurological complications | 4 (7.14%) | 11 (18.64%) | 2.413 | 0.12 |

**Table 4 Correlation analysis of DII with disease severity and complications in children with HSP.**

| Parameters | r | $R^2$ | *p* |
|---|---|---|---|
| Renal complications | 0.21 | 0.044 | 0.024 |
| Skin complications | 0.243 | 0.059 | 0.009 |
| Gastrointestinal complications | 0.276 | 0.076 | 0.003 |
| Coagulation disorders | 0.286 | 0.082 | 0.002 |
| Respiratory complications | 0.241 | 0.058 | 0.01 |
| Neurological complications | 0.171 | 0.029 | 0.068 |

the potential influence of dietary inflammatory potential on disease outcomes in pediatric patients.

## DISCUSSION

The findings of this study shed light on the potential influence of dietary factors, specifically the DII, on disease severity and complications in children diagnosed with HSP. The results revealed significant associations between higher DII and elevated inflammatory biomarkers, adverse nutrient intake, blood lipid profiles and disease complications. These findings provide valuable insights into the interplay between dietary patterns and inflammatory responses in pediatric vasculitis, specifically HSP, and have implications for the management and potential dietary-based interventions in HSP.

First of all, the study observed consistent and significant elevations in inflammatory biomarkers among participants in the Pro-inflammatory dietary group compared to those in the anti-inflammatory dietary group. These biomarkers included C-reactive protein (CRP), tumor necrosis factor-α (TNF-α), interleukin-6 (IL-6), erythrocyte sedimentation rate (ESR), white blood cell count, eosinophils, and IgE. The higher levels of these inflammatory markers in the pro-inflammatory dietary group suggest a pro-inflammatory state associated with dietary patterns characterized by a higher DII score. These findings are in line with previous research of *de Mello et al. (2023)* indicating that dietary patterns with a high DII score tend to contribute to systemic inflammation, which may predispose individuals to various inflammatory conditions and potentially exacerbate existing inflammatory diseases.

In addition to the inflammatory biomarkers, the study also identified significant differences in macronutrient intake and blood lipid profiles between the low and pro-inflammatory dietary groups. Participants in the pro-inflammatory dietary group exhibited higher consumption of total calories, protein, carbohydrates, fat, and fiber, which were associated with a more pro-inflammatory diet. Moreover, the Pro-inflammatory dietary group demonstrated adverse lipid profiles, including elevated levels of total cholesterol, LDL cholesterol, HDL cholesterol, triglycerides, and VLDL cholesterol. These findings highlight the potential impact of dietary inflammatory potential on lipid metabolism and suggest a link between DII and metabolic health in the context of pediatric vasculitis (*Mohammadi et al., 2023*). Given the established relationship between dyslipidemia and cardiovascular risk, the observed adverse lipid profiles in the pro-inflammatory dietary group raise concerns about the long-term cardiovascular implications of dietary patterns with a higher inflammatory potential in children with HSP (*Xu, Li & Wu, 2022*).

Furthermore, the study revealed a notable association between high DII and an increased prevalence of disease complications, including renal, skin, gastrointestinal, coagulation disorders, and respiratory complications. These findings underscore the potential impact of dietary inflammatory potential on the development of complications in children with HSP. This suggests that dietary patterns with a higher inflammatory potential may contribute to disease progression and severity, emphasizing the need to consider dietary interventions as part of the holistic management of pediatric vasculitis.

The observed correlations between DII and various complications, have important clinical implications for the management of HSP in pediatric patients. Understanding the potential impact of dietary patterns on disease outcomes can provide healthcare professionals with valuable insights for developing tailored dietary interventions to complement traditional treatment approaches. Incorporating dietary assessments and interventions aimed at modulating the inflammatory potential of the diet could be beneficial in mitigating disease severity and reducing the risk of complications in children with HSP.

While the study provides important insights into the association between DII and disease severity and complications in children with HSP, several limitations should be acknowledged. The retrospective nature of the cohort study introduces inherent biases that are typical of such designs. Recall bias may have influenced the accuracy of dietary recall, particularly since dietary habits were assessed retrospectively. Prospective studies would be better suited to capture real-time dietary patterns and their dynamic changes over time. Addition, the study's dependence on self-reported dietary data, collected through the Food Frequency Questionnaire (FFQ), may have introduced potential biases. The responses of the participants could have been affected by memory limitations, social desirability bias, or insufficient knowledge of portion sizes, potentially leading to under- or over-reporting of food intake. These factors may have impacted the accuracy of the dietary data collected. This limitation underscores the need for future studies to incorporate multiple methods of dietary assessment, such as food diaries or biomarkers, to triangulate dietary intake data and improve the accuracy of dietary intake estimation. Moreover, the use of a single dietary

assessment tool, the FFQ, to quantify dietary inflammatory potential may not have captured the full complexity of dietary patterns and their temporal variability. Longitudinal studies utilizing multiple dietary assessment tools at different time points could provide a more comprehensive picture of dietary habits and their impact on health outcomes. The findings of this study may not be directly applicable to other populations or ethnic groups due to potential differences in dietary habits and genetic backgrounds. Future research should aim to replicate these findings in diverse populations to enhance the generalizability of the results.

## CONCLUSION

In conclusion, the findings of this study underscore the potential influence of dietary inflammatory potential, as quantified by the DII, on disease severity and complications in children diagnosed with HSP. The observed associations between higher DII and elevated inflammatory biomarkers, adverse nutrient intake, adverse lipid profiles and disease complications highlight the relevance of considering dietary patterns in the management of pediatric vasculitis. Moving forward, integrating dietary assessments and interventions as part of a comprehensive approach to managing pediatric vasculitis, including HSP, has the potential to optimize clinical outcomes and improve the overall well-being of affected children. Further research and multidisciplinary collaborations are warranted to explore the role of dietary interventions in mitigating disease severity and reducing the risk of complications in pediatric patients with vasculitis.

### Funding

This study was supported by the Excellent Talent Team of Hainan Province (No. QRCBT202121), the Hainan Province Clinical Medical Center (No. QWYH202175), and the Natural Science Foundation of Hainan Province (No. 821RC1130). The funders had no role in study design, data collection and analysis, decision to publish, or preparation of the manuscript.

### Grant Disclosures

The following grant information was disclosed by the authors:
Excellent Talent Team of Hainan Province: QRCBT202121.
Hainan Province Clinical Medical Center: WYH202175.
Natural Science Foundation of Hainan Province: 821RC1130.

### Competing Interests

The authors declare that they have no competing interests.

### Author Contributions

- Jinshu Chen conceived and designed the experiments, authored or reviewed drafts of the article, and approved the final draft.

- Pihou Chen performed the experiments, prepared figures and/or tables, and approved the final draft.
- Yijin Song analyzed the data, prepared figures and/or tables, and approved the final draft.
- Jiaxin Wei performed the experiments, prepared figures and/or tables, and approved the final draft.
- Shiya Wu analyzed the data, prepared figures and/or tables, and approved the final draft.
- Fan Wu performed the experiments, prepared figures and/or tables, and approved the final draft.
- Zhiquan Xu conceived and designed the experiments, authored or reviewed drafts of the article, and approved the final draft.

## Human Ethics

The following information was supplied relating to ethical approvals (*i.e.*, approving body and any reference numbers):

This study was approved by the Ethics Committee of Hainan Women and Children's Medical Center and complies with the ethical guidelines of the Helsinki Declaration.

## Data Availability

The original data is available in the Supplemental File.

## Supplemental Information

Supplemental information for this article can be found online at http://dx.doi.org/10.7717/peerj.18175#supplemental-information.

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
