# Peer review of "The relationship between the severity and complications of Henöch-Schönlein purpura in children and dietary inflammatory index: a retrospective cohort study"

_PeerJ, doi:10.7717/peerj.18175_

## Round 0.1 · original submission · Major Revisions

1. More details are needed on the blood lipid profile measurements. The authors should specify the assays/instruments used to measure total cholesterol, LDL, HDL, triglycerides, and VLDL. The protocols for blood sample collection, processing, and analysis should also be described.
2. Information is lacking on the fasting requirements and blood draw procedures for collecting the samples used to measure lipid levels and inflammatory biomarkers.
3. The authors need to discuss the quality control measures implemented to ensure the accuracy and precision of the lipid and inflammatory biomarker measurements.
4. The method for quantitatively assessing HSP severity based on purpura distribution, joint involvement, etc. is not clearly defined. More details are required on how the percentages for mild, moderate and severe HSP were determined.
5. There is unnecessary repetition of information between the Methods and Results sections. For example, the sentence "A retrospective cohort study was conducted, enrolling children aged 2-14 years diagnosed with HSP" appears verbatim in both sections. This redundancy should be eliminated.
6. The detailed statistical analysis methods described in the Results section (e.g. chi-square test, t-test, correction formulas) are already covered in the Methods. This repetition should be removed.
7. The presentation of results in each subsection tends to be repetitive, often reiterating that the Pro-inflammatory diet group had worse outcomes compared to the Anti-inflammatory diet group. Consider consolidating this information to improve clarity and reduce redundancy.
8. The Results reiterate basic information about the study groups and their demographic/clinical characteristics multiple times without adding significant new insights. Repeated phrases like "Participants divided into low and Pro-inflammatory dietary groups based on their DII scores" can be removed.
9. The limitations of the study, including its retrospective design, reliance on self-reported dietary data, and use of a single dietary assessment should be expanded.
10. Future directions, such as the need for prospective longitudinal studies with larger sample sizes and more comprehensive dietary assessments, should be discussed in more detail.

·

Basic reporting

The study is commendable for its thorough data collection and statistical analysis, contributing valuable insights into the role of dietary inflammation in pediatric HSP. However, to enhance clarity and robustness, it would benefit from addressing some redundancies and ensuring consistency in presenting methodological details, particularly regarding blood lipid measurements, data transformation techniques, and detailed dietary assessment procedures. By addressing these anomalies, the manuscript can present a more coherent, non-redundant, and analytically rigorous discussion of the study’s significant findings and implications regarding dietary intake, inflammation, and disease complications in children with HSP.

Experimental design

1. There is no explicit mention in the "Methods" section about how the blood lipid profiles were measured and analyzed. While it discusses general data collection from medical records and assessments, there are no specific details provided for laboratory procedures or tools used to measure lipid levels, such as: The specific assays or instruments used for measuring total cholesterol, LDL, HDL, triglycerides, and VLDL. The protocols followed for blood sample collection, processing, and analysis.
2. There is no description of the procedure for blood sample collection (e.g., fasting requirements, blood draw protocol).
3. There is no mention of the quality control measures taken to ensure the accuracy and precision of the lipid measurements.

Validity of the findings

1. Addressing above gaps would enhance the transparency and reproducibility of the study, allowing readers to better evaluate the robustness of the findings related to blood parameters.
2. There is no clear tie-back to how the specifics of purpura distribution, joint involvement, etc., directly translate into the percentages given for severity. The detail of how severity is quantitatively assessed is missing.

Additional comments

1. Repeated Information in the "Methods" and "Results" and Redundancy in Presentation. The "Methods" section states: "A retrospective cohort study was conducted, enrolling children aged 2-14 years diagnosed with HSP." The "Results" section starts with the exact same phrase: "A retrospective cohort study was conducted, enrolling children aged 2-14 years diagnosed with HSP." This exact repetition is unnecessary and should be avoided. Each section should be unique and not repeat information verbatim.
2. Detailed Redundancy in "General Information" and "Statistical Analysis". The "Methods" section includes detailed instructions on statistical tests, such as the chi-square test, t-test, and correction formulas. This is thorough but repeats the statistical significance description in the "Results":“The demographic and clinical characteristics...no significant differences in demographic and clinical characteristics between the low DII and Pro-inflammatory dietary groups, indicating a well-balanced study population.” That is, author should present demographic and clinical characteristics concisely without unnecessarily repeating cohort properties.
3. The presentation of findings in each subsection of "Results" tends to be repetitive. For example, the results for inflammatory biomarkers, nutrient intake, blood lipid profile, disease complications, and disease severity often reiterate similar patterns—linking higher DII to adverse outcomes. Consolidating this information might provide better clarity and avoid redundancy. Specifically, phrases like: "...participants in the Pro-inflammatory dietary group consistently demonstrated significantly elevated levels...” “…the Anti-inflammatory dietary group had a higher percentage of participants with mild disease severity...”
4. The demographic and clinical characteristics characterization is provided multiple times across methods and results without adding substantial new insight in the latter sections. “Participants…divided into low and Pro-inflammatory dietary groups based on their DII scores” is repetitively mentioned.

Reviewer 2 ·

Basic reporting

The article is clearly written which appropriate literature references. The article is well-structured and easy-to-follow.

Experimental design

The research question is well defined. Please see additional comments for specific questions.

Validity of the findings

The statistical analysis employed is reasonable and appropriate. Please see additional comments for specific questions.

Additional comments

Chen et al. described a retrospective study to investigate the relationship between dietary inflammation potential and HSP severity in HSP children. Based on the children’s dietary intake as measured by Dietary Inflammatory Index (DII), the authors separated the children into anti-inflammatory potential and pro-inflammatory diet groups. The authors found that the pro-inflammatory diet group showed higher levels of inflammatory markers, different dietary patterns, and higher blood lipid levels. In addition, more children in the pro-inflammatory diet group have more disease complications and higher HSP severity.
Overall, this study was clear and well-structured. The authors provided the data underlying the analysis and figures. The statistical method used was appropriate and was described in detail in the methods section. I have the following comments:
1. The “Results” section in the abstract describes the method used. Instead, the authors should summarize the findings from these methods.
2. In Materials and Methods Section 2.2, the authors assigned positive, zero, and negative values to indicate pro-inflammatory, anti-inflammatory and non-inflammatory effect. Later in Section 2.5, the authors stated that “a negative DII indicating anti-inflammatory potential and a positive DII indicating pro-inflammatory potential.” Could they please clarify the inconsistency?
3. In Table 2, the authors described biomarkers that are elevated in the pro-inflammatory groups. How are these biomarkers chosen? They did observe any anti-inflammatory molecules that were suppressed in the pro-inflammatory group? For the molecules known to not have effects on HSP/inflammation, do they observe no significant changes? (i.e. Are the “negative controls” behaving as expected based on this statistical analysis?)
4. It would help to clarify the description of DII if the authors can provide an example input and the associated values on how it was calculated.
5. It would help reproducibility if the DII information is provided in the supplementary table.

---

## Round 0.2 · accepted · Accept

Since all comments have been addressed by authors, I have no inquiries. I agree the publication of this manuscript.

·

Basic reporting

Authors addressed and answered all my comments

Experimental design

NG

Validity of the findings

NG

Additional comments

NG

Reviewer 2 ·

Basic reporting

No comment

Experimental design

The authors added useful details to describe the experimental and statistical methods used.

Validity of the findings

no comment